# Protein Plasma Levels of the IGF Signalling System Are Altered in Major Depressive Disorder

**DOI:** 10.3390/ijms242015254

**Published:** 2023-10-17

**Authors:** Carlos Fernández-Pereira, Maria Aránzazu Penedo, Tania Rivera-Baltanás, Tania Pérez-Márquez, Marta Alves-Villar, Rafael Fernández-Martínez, César Veiga, Ángel Salgado-Barreira, José María Prieto-González, Saida Ortolano, José Manuel Olivares, Roberto Carlos Agís-Balboa

**Affiliations:** 1Translational Neuroscience Group, Galicia Sur Health Research Institute (IIS Galicia Sur), Área Sanitaria de Vigo-Hospital Álvaro Cunqueiro, SERGAS-UVIGO, CIBERSAM-ISCIII, 36213 Vigo, Spain; carlosfernandezpereira@gmail.com (C.F.-P.); aranchapenedo@gmail.com (M.A.P.);; 2Neuro Epigenetics Lab, Health Research Institute of Santiago de Compostela (IDIS), Santiago University Hospital Complex, 15706 Santiago de Compostela, Spain; josemaoscar.prieto@usc.es; 3Rare Disease and Pediatric Medicine Group, Galicia Sur Health Research Institute (IIS Galicia Sur), SERGAS-UVIGO, 36312 Vigo, Spain; tania.perez@iisgaliciasur.es (T.P.-M.); marta.alves@iisgaliciasur.es (M.A.-V.); saida.ortolano@iisgaliciasur.es (S.O.); 4Cardiovascular Research Group, Galicia Sur Health Research Institute (IIS Galicia Sur), 36213 Vigo, Spain; 5Department of Preventive Medicine and Public Health, Health Research Institute of Santiago de Compostela (IDIS), University of Santiago de Compostela, 15706 Santiago de Compostela, Spain; 6Consortium for Biomedical Research in Epidemiology and Public Health (CIBER en Epidemiología y Salud Pública-CIBERESP, 28029 Madrid, Spain; 7Translational Research in Neurological Diseases Group, Health Research Institute of Santiago de Compostela (IDIS), Santiago University Hospital Complex, SERGAS-USC, 15706 Santiago de Compostela, Spain; 8Neurology Service, Santiago University Hospital Complex, 15706 Santiago de Compostela, Spain

**Keywords:** Insulin-like growth factor 2 (IGF-2), Insulin-like growth factor binding protein 1 (IGFBP-1), Insulin-like growth factor binding protein 3 (IGFBP-3), Insulin-like growth factor binding protein 5 (IGFBP-5), Insulin-like growth factor binding protein 7 (IGFBP-7), depression, Hamilton Depression Rating Scale (HDRS), Mini-Mental State Examination (MMSE), Free and Cued Selective Remaining Test (FCSRT), Self-Assessment Anhedonia Scale (SAAS)

## Abstract

The Insulin-like growth factor 2 (IGF-2) has been recently proven to alleviate depressive-like behaviors in both rats and mice models. However, its potential role as a peripheral biomarker has not been evaluated in depression. To do this, we measured plasma IGF-2 and other members of the IGF family such as Binding Proteins (IGFBP-1, IGFBP-3, IGFBP-5 and IGFBP-7) in a depressed group of patients (n = 51) and in a healthy control group (n = 48). In some of these patients (n = 15), we measured these proteins after a period (19 ± 6 days) of treatment with antidepressants. The Hamilton Depressive Rating Scale (HDRS) and the Self-Assessment Anhedonia Scale (SAAS) were used to measure depression severity and anhedonia, respectively. The general cognition state was assessed by the Mini-Mental State Examination (MMSE) test and memory with the Free and Cued Selective Reminding Test (FCSRT). The levels of both IGF-2 and IGFBP-7 were found to be significantly increased in the depressed group; however, only IGF-2 remained significantly elevated after correction by age and sex. On the other hand, the levels of IGF-2, IGFBP-3 and IGFBP-5 were significantly decreased after treatment, whereas only IGFBP-7 was significantly increased. Therefore, peripheral changes in the IGF family and their response to antidepressants might represent alterations at the brain level in depression.

## 1. Introduction

Depression is a chronic medical illness that affects thoughts, mood, and physical health [1]. Depression is one of the most common psychiatric disorders affecting around 5% of the worldwide population [2]. Recently, the prevalence of depression and anxiety disorders increased by 25% due to the COVID-19 pandemic, according to the World Health Organization (WHO) [2,3]. Approximately, only one-third of patients treated for depressive disorders will reach the remission criteria [4]. Understanding inner vulnerabilities that may predispose to suffer depression, such as studying peripheral biomarkers, could be of particular interest for comprehending the etiology of depression, and may be crucial to facilitate a more specific, rapid, and successful treatment [5].

The Insulin-like growth factor (IGF) system is formed from two ligands, IGF-1 and IGF-2, that regulate multiple physiological processes in mammalian development, metabolism, and growth [6]. These two ligands act via the insulin receptor variants (IR-A and IR-B), the type I Insulin-like growth factor receptor (IGF-1R) that are tyrosine kinase receptors [7]. Upon receptor binding, the intracellular tyrosine kinase domain autophosphorylation activates two main signaling pathways: the phosphoinositide 3-kinase (PI3K)-Akt/protein kinase B (PKB) and the Ras-mitogen-activated protein kinase (MAPK) pathways [8]. The biological effects of the activation of these pathways are cell growth, cell differentiation, cell survival, proliferation, and migration [9]. However, the main cognitive and neuroprotective effects of the IGF-2 signaling seem to be mediated by its effects via IGF-2R [10].

On the other hand, IGF activity is regulated by the IGF-binding proteins (IGFBPs 1 to 7) that bind both IGF-1 and IGF-2 with high affinity, but not insulin [11]. These binding proteins inhibit IGF actions by preventing their union with their respective receptors, or can potentiate their actions by strengthening the ligand–receptor interaction. In general terms, it is assumed that IGFBPs prolong the circulating half-life of IGFs throughout the regulation of their movement across tissues [12]. However, IGFBPs also display IGF-independent actions by directly interacting with cell surface receptors. What is more, IGFBPs can also enter the nucleus via endocytosis and modulate the transcription process by their binding to nuclear receptors [13].

IGF-2 is the most abundantly expressed IGF ligand in the adult central nervous system (CNS) [14], and it plays a key role via IGF–1R signaling in nerve growth during development by regulating the growth survival, maturation, and proliferation of different types of nerves cells such as oligodendrocytes, nerve precursor cells, astrocytes, or nerve stem cells [15]. The gene expression of *igf2* has been found to be downregulated in the central nucleus of the amygdala [16] and hippocampus [17] in depressed rat models. Conversely, in a validated mice model of anhedonia, the *igf2* hippocampal gene expression was found to be non-significant between non-treated mice and controls [18]. Nonetheless, it has been recently shown that IGF-2 intrahippocampal injections not only alleviate depression-like behaviors in both rat [19] and mice [20] models, but also enhance memory consolidation in rats [21]. In humans, the variable methylation of the IGF-2 gene has been found to be related with the clinical psychopathological condition of depression in monozygotic twins [22], with maternal anxiety or depressive behavior during pregnancy being correlated with decreased DNA methylation in the IGF-2 gene in the offspring at birth [23,24] and related to low weight at birth [25,26], which might be a risk factor for developing anxiety or depression at adulthood [27]; nonetheless, it still remains a matter of debate [28].

In the context of psychiatric disorders, IGF-2 peripheral levels have been previously studied in schizophrenia [29,30,31,32] and in neurodegenerative disorders that impair memory or cognitive status, such as Alzheimer’s disease [33,34,35], Huntington’s disease [36] and Parkinson’s disease [37]. However, to the best of our knowledge, there is no literature available on IGF-2 peripheral levels in major depressive disorders involving human samples. In contrast, peripheral IGF-1 has received far more attention in the last few decades, not just in depression [38,39,40,41,42,43,44,45,46,47,48,49], but also in other mental conditions such as schizophrenia [29,49,50,51,52,53,54,55,56] and bipolar disorder [57,58,59,60,61,62]. On the other hand, circulating IGFBPs have been less widely studied than their ligand counterparts in the field of psychiatry, and always in a context of their IGF-dependent actions [29,30,31,32,38,55,63,64,65]. Some IGFBPs have received more attention than others. In detail, we did not find any literature on the peripheral levels of IGFBP-4, IGFBP-5 or IGFBP-6 in psychiatric disorders. On the contrary, IGFBP-1 [63,64], IGFBP-2 [38,64,65] and IGFBP-3 [29,30,38,55] have been more widely studied in this context.

Researchers from our group have previously studied the roles of both IGF-2 and IGFBP-7 in schizophrenia [32], as well as in the extinction of fear memories in mouse models [66,67] and as possible targets for treatment in Alzheimer’s disease [68]. Given that IGF-2 can trigger IGF-1R signaling and IGF-2R to act as a memory enhancer, we believe there are reasons to study the possible roles of peripheral levels of IGF-2 and some IGFBPs in the context of depression. Therefore, our aim is to check whether plasma levels of IGF-2, IGFBP-1, IGFBP-3, IGFBP-5 and IGFBP-7 might reflect changes between patients and controls, and also whether treatment with antidepressants (AD) may recover these levels and if they correlate with changes in subjective scales that measure depression severity (HDRS), anhedonia (SAAS), general cognitive state (MMSE) and memory (FCSRT).

## 2. Results

### 2.1. General Data and Subjective Scales

Demographic and clinical data are reported in Table 1. The two groups (C, D) were not significantly different in terms of gender distribution (χ^2^ (1, n = 99) = 1.67, *p* = 0.230), nor in median age (F (1) = 2.67, *p* = 0.153). On the other hand, the SAAS score has been found to be significantly higher in depressed patients when compared to controls (U = 20, *p* < 0.001) (Table 1).

In Table 2, we show the characteristics of the group of patients diagnosed with depression before and after AD treatment. The Wilcoxon matched pairs signed rank test showed that the SAAS scale was not significantly changed after treatment in patients with depression (W = −40, *p* = 0.277), whereas the HDRS was significantly reduced after treatment (*t* (14) = 9.28, *p* < 0.001), as it was the MMSE test (*t* (14) = 2.20, *p* < 0.05) (Table 2). In the case of the FCSRT, the four subitems FCSRT_TFR (*t* (14) = 3.84, *p* < 0.01), FCSRT_TR (W = 77, *p* < 0.05), FCSRT_DFR (W = 77, *p* < 0.01) and FCSRT_DTR (W = 72, *p* < 0.01) were found to be significantly increased after treatment in depressed patients (Table 2).

### 2.2. Correlation between Plasma IGFs Proteins and Metabolic Parameters

All mean values of the metabolic parameters in the depressed group of patients (D) were in the physiological range, indicated in the first brackets: glucose (85.75 ± 17.26 mg/dL) (73–100 mg/dL), albumin (3.92 ± 0.41 g/dL) (3.4–5 g/dL), cholesterol (170.46 ± 31.78 mg/dL) (100–200 mg/dL) and triglycerides (116.90 ± 73.95 mg/dL) (50–150 mg/dL).

No significant correlation has been found between any IGF proteins measured in this study and the four metabolic parameters glucose, albumin, cholesterol, and triglyceride blood levels in the cohort of depressed patients (Table 3). We show the exact *p*-values in Table 3.

### 2.3. Levels of IGF Proteins in Depressed Patients and Healthy Controls

The levels of IGF-2, IGFBP-1, IGFBP-3, IGFBP-5 and IGFBP-7 were compared between the control group and the depressed group (Table 1).

The levels of both IGF-2 (U = 197, *p* < 0.001) and IGFBP-7 (*t* (97) = 3.061, *p* < 0.01) were significantly increased in the depressed group of patients when compared against controls (Figure 1a,e). However, this was not the case for IGFBP-1, IGFBP-3 and IGFBP-5, which remained statistically undifferentiated despite showing an increased tendency in the depressed group (Figure 1b–d).

However, overall age was found to be significantly correlated with IGF-2 (r_s_ (98) = 0.342, *p* = 0.001), IGFBP-1 (r_s_ (99) = 0.342, *p* = 0.001) and IGFBP-7 (r_p_ (99) = 0.422, *p* = 1.34 × 10^−5^). In the control group, age was still significantly correlated with IGFBP-1 (r_s_ (99) = 0.597, *p* = 1.31 × 10^−5^), while in the depressed group, age only correlated with IGFBP-7 (r_p_ (99) = 0.451, *p* = 0.001). Moreover, the overall median IGFBP-3 levels were found to be significantly higher in women than in men (women = 547.4 ng/mL, n = 30; men = 438.3 ng/mL, n = 21, U = 162, *p* = 0.0029). The same was observed for IGFBP-5 (women = 90.81 ng/mL, n = 30; men = 78.15 ng/mL, n = 21, U = 189.5, *p* = 0.0156). No other correlation or significant difference was found regarding IGFs proteins with age or gender, respectively.

Subsequently, we performed a linear regression analysis to verify whether the group variable (either control or depression condition) could predict significantly different values of the dependent variables (IGF proteins) regardless of other independent variables in the model, such as age and gender. Specifically, we evaluated IGF-2 and IGFBP-7, since they were both significantly altered between controls and patients.

Therefore, we calculated different linear regression models using the backpropagation methodology to predict the effects of group (control, depression), age (years) and gender (male, female) over the IGF-2 and IGFBP-7 levels. In the case of IGF-2, both age and gender were eliminated in model 3 (Table 4). The regression equation was statistically significant (F (1, 96) = 49.96, *p* < 0.001), R^2^ = 0.34, which indicates that 34% of the change in IGF-2 levels can be explained by model 3 with the variable group, excluding other independent variables such as age and gender (Table 4).

Nonetheless, model 2 eliminated the variable of group for IGFBP-7 (Table 5), meaning that the differences we observed in IGFBP-7 levels can be better explained by variables such as age and gender than group.

### 2.4. The Levels of IGF Proteins in Depressed Patients before and after Treatment

The levels of IGF-2 (W = −102, *p* < 0.05), IGFBP-3 (W = −116, *p* < 0.001) and IGFBP-5 (W = −120, *p* < 0.001) were significantly decreased in depressed patients after treatment (Figure 2a,c,d), whereas only IGFBP-7 (W = 88, *p* < 0.05) was significantly increased (Figure 2e). In the case of IGFBP-1, no statistical difference was found between before and after AD treatment (Figure 2b) (Table 2).

### 2.5. Correlation between Plasma IGFs Proteins and Subjective Scales

In the depressed group of patients that took part in the longitudinal study, IGFBP-3 and IGFBP-5 plasma protein levels positively correlated with the MMSE test ((r_s_ (14) = 0714, *p* = 0.003) and (r_s_ (14) = 0.670, *p* = 0.006), respectively) and with the Total Recall subitem in the FCSRT_TR ((r_s_ (14) = 0.754, *p* = 0.001) and (r_s_ (14) = 0.672, *p* = 0.006)) before treatment (D0). What is more, IGFBP-3 was also found to be significantly correlated with the Total Free Recall subitem in the FCSRT_TFR (r_s_ (14) = 0.556, *p* = 0.031) (Table 6).

Intriguingly, only IGFBP-5 levels remained positively correlated with the MMSE test (r_s_ (14) = 0.66, *p* = 0.007), the FCSRT_TFR (r_s_ (14) = 0.608, *p* = 0.016) and the FCSRT _TR (r_s_ (14) = 0.566, *p* = 0.028) after treatment. On the contrary, IGFBP-7 levels negatively correlated with the MMSE test (r_s_ (14) = −0.287, *p* = 0.015) after treatment (Table 7).

## 3. Discussion

To the best of our knowledge, this is the first time that IGF-2, IGFBP-5 and IGFBP-7 plasma protein levels have been addressed in depressed patients. On the contrary, in the last few decades, IGF-1 has received far more attention than IGF-2 in the context of depression [38,39,40,41,42,43,44,45,46,47,48,49]. In some studies, IGF-1 peripheral levels were found to be significantly increased in depression [38,39,40,43,44,49] regardless of ethnicity, since some works were done in Europe [38,39,40,43], North America [44] and Japan [49]. What is more, IGF-1 was found upregulated either when patients were following AD treatments [40,43,49] or not [38,39,44] by the time protein measurements were taken. This may suggest that the levels of IGF-1 might be increased in depression regardless of treatment conditions.

Nevertheless, in other studies, no significant differences were found in the levels of IGF-1 between depressed patients and controls [45,46,47,48]. Again, these studies have been undertaken on different ethnicities such as European [46,48], Chinese [45] and North American [47], although only on men in the last region. In some studies, patients were following AD treatments [47,48], or were drug-free in another [46]. More interestingly, in one study, patients were completely drug-naïve [45]. Conversely, in the work of Bot et al., in 2016, it was found that levels of peripheral IGF-1 were significantly increased in depressive patients under AD treatment, but significantly decreased in patients that were non-responders, and who were not following AD treatment [41].

Therefore, cross-sectional studies have primarily been made to compare peripheral IGF-1 levels between patients and controls, which may indicate the potential role of these levels as a diagnostic or predictor biomarker. However, some longitudinal studies addressed changes in peripheral IGF-1 in response to AD treatment [38,40,42,43,45,46]. The majority found that peripheral IGF-1 levels tend to significantly decrease after a period on AD, regardless of ethnicity, drug type, dosage, or regime [38,42,43,45]. To some extent, these studies make a distinction between patients who respond (remitters) to AD treatment and those who do not (non-remitters), normally reflected in a reduction by 50% in the HDRS scale [69]. This aspect may be important in establishing peripheral IGF-1 or IGF-2 levels as objective distinguishing biomarkers that can reflect changes in subjective scales of clinical use, such as the Hamilton Depression Rating Scale (HDRS) in the case of depression. Specifically, in the work of Kopzack et al., 2015, it was found that after 6 months of treatment with escitalopram, no significant difference was found between responders and non-responders. However, the levels of IGF-1 were significantly higher in non-responders at baseline [40]. To some extent, this result must be carefully interpreted, since those patients were also under treatment by the time they were included in the study [40]. Another study found no difference between before and after AD treatment in IGF-1 peripheral levels [46].

In our case, we did not measure IGF-1, but we did measure IGF-2, since peripheral IGF-2 has never been measured in the context of human depression. From our perspective, the study of IGF-2 is interesting in the context of psychiatric disorders because IGF-2 can trigger IGF-1R signaling [9], as well as IGF-2R signaling [10], whereby IGF-2 binding may act as a memory enhancer [70] and a pro-cognitive agent through the induction of neurogenesis [71]. In the present study, we found that the levels of IGF-2 were statistically increased in depression when compared to controls, regardless of age and sex. Moreover, we found that after 19 days of AD treatment, levels of IGF-2 significantly decreased in depressed patients. Our results in IGF-2 follow the same tendency that previous studies found in IGF-1. 

On the other hand, peripheral levels of IGFBPs have primarily been studied in an IGF-dependent manner in psychiatric disorders. IGFBPs display multiple functions related to the functional regulation of IGFs (IGF-dependent actions), such as the transport of IGFs through plasma, as well as the regulation of the efflux of IGFs ligands from the vascular space and their clearance [72]. Moreover, IGFBPs control the tissue-specific direction of IGFs ligands and the regulation of the interaction of IGFs with their respective receptors [73,74]. However, given the fact that IGFBPs can also exert IGF-independent actions [75], it might also be interesting to consider IGFBPs as possible separated biomarkers in the context of psychiatric disorders. Previous studies have compared the proteomic profiles of depressed patients and healthy controls, finding that in atypical depression, but not melancholic, both IGFBP-1 and IGFBP-2 were significantly decreased [64]. Interestingly, other studies found no significant differences in serum IGFBP-2 levels between depressed patients and healthy controls [38,65]. IGBFP-2 levels did not significantly differ between treated and drug-free patients [65]. On the other hand, IGFBP-3 shows the characteristic of being the most abundant among IGFBPs in circulation [76], and transports IGFs by forming a ternary complex with an acid-labile subunit (ALS) [77]. The ternary complex is responsible for the transport of almost 80% of IGFs in circulation, while the rest of the IGFBPs are responsible for the transport of the remaining 20%, which are transported in binary complexes [72]. IGFBP-3 levels have been found non-significantly between patients and controls, as well as between patients before and after AD treatment [38]. Finally, we have found no study addressing peripheral IGFBP-4, IGFBP-5 and IGFBP-6 levels in the context of human depression. In our study, we found that IGFBP-1, IGFBP-3 and IGFBP-5 did not significantly differ between patients and controls. However, the levels of IGFBP-3 and IGFBP-5 were significantly reduced in depressed patients after AD treatment, as it was the case of their ligand IGF-2. These results might point to a similar mechanism of regulation.

Conversely, IGFBP-7 has less affinity for IGFs than the other six IGFBPs (1-6) [78], and there is no previous literature addressing the plasma levels of IGFBP-7 in depression. Nonetheless, IGFBP-7 could be a promising biomarker in the context of depression, since IGFBP-7 has been proposed as a potential therapeutic target in Alzheimer’s disease [68], for which depression is a risk factor [79]. Moreover, plasma IGFBP-7 levels have been found to be significantly decreased in schizophrenic patients [30]. On the contrary, in our previous work, we found that plasma IGFBP-7 levels were significantly increased in schizophrenic patients [32]. According to Yang et al., 2020 [30], the disparity observed in the results among different studies on schizophrenia (SZ) could be related to several aspects, such as long-term treatment conditioning, ethnicity, intrinsic SZ heterogeneity, and some other potential confounding factors that could be altering IGF levels, such as albumin. These aspects should also be considered when studying the presence of IGF proteins in other mental conditions, such as depression. In the present study, the levels of IGFBP-7 were initially found to be significantly increased in depressed patients, but after correcting for age and gender, we found that depression might not explain this difference. On the other hand, IGFBP-7 was the only IGFBP protein that was significantly increased after treatment. One possible explanation is that since IGFBP-7 binds IGF ligands with less affinity that the other six IGFBPs [78], the regulation of IGFBP-7 in response to treatment with AD might have a different mechanism.

In psychiatry, one key aspect of a potential biomarker is that its variations can be correlated with changes in the subjective scales used to help clinicians with diagnosis, prognosis, monitoring, or response to treatment, in turn helping them to develop more specialized and individualized medicine [80,81,82]. In this context, IGF-1 has been assessed for its correlation with subjective scales of depression [39,40,43,45,46,49]. For the most part, no significant correlation has been found between IGF-1 and the Hamilton Depression Rating Scale (HDRS) [39,40,45,46]. However, one study found a positive significant correlation between IGF-1 peripheral levels and the HDRS scale [49]. In our present study, we did not find any significant correlation between IGF proteins and the HDRS.

On the other hand, anhedonia has been historically conceptualized as a “loss of pleasure”, but recent research has revealed different aspects, which include hedonic function, desire, effort, motivation, anticipation and consummatory pleasure, and these have been included in the DSM-V as a core feature of depression [83]. However, we did not find any correlation between the SAAS scale and any peripheral IGF member. Changes in these proteins might not manifest the differences observed in SAAS between either controls and patients or before and after AD treatment. In summary, we did not find any correlation between the IGFs proteins and the score on a subjective scale that is used to assess depression symptomatology.

However, in terms of general cognition, we have found a statistically positive correlation between the MMSE test and both IGFBP-3 and IGFBP-5. In the case of IGFBP-5, this correlation persisted even after treatment. In depression, it has been observed that a mild cognitive impairment might represent a risk factor for developing Alzheimer’s disease. Moreover, the relationship between IGF-1 and cognitive impairment has been widely studied in Alzheimer´s disease [34,35,84,85,86,87,88,89,90], but offered discrepant results. Nonetheless, according to the neurotrophic hypothesis, depression could be the result of the stress-induced reduced expression of neurotrophic factors such as Brain-Derived Neurotrophic Factor (BDNF) [91], IGF-1 or Glial cell line-derived neurotrophic factor (GDNF) [92]. Baring this in mind, the positive correlation between peripheral IGF family members and cognition opens an interesting field in the context of depression that should be explored more deeply. On the other hand, IGBFP-7 is the only protein that was negatively correlated with the MMSE after treatment.

Interestingly, episodic memory, evaluated through the Free and Selective Cued Reminding Test (FCSRT), has been found to be correlated with some IGFBPs members in the context of depression. Specifically, IGFBP-3 levels were positively correlated with both Total Free Recall and Total Recall subitems before treatment, but these correlations were lost after treatment. However, this was not the case for IGFBP-5, which maintained its statistically positive correlation both before and after treatment with Total Free Recall. The FCSRT was created to coordinate both acquisition and retrieval using the same semantic cues to control learning and elicit effective cued recall, enabling encoding specificity [93]. Encoding specificity is a tool that enables efficient learning and memory. However, in neurodegenerative states such as Alzheimer´s disease, some brain regions that are essential to these processes are impaired. Thus, tests that maximize the process of encoding specificity, such as the FCSRT, may be sensitive enough to detect early Alzheimer´s disease or Mild Cognitive Impairment (MCI) [94,95]. However, the FCSRT was used in our study to assess whether episodic memory could be improved after AD treatment in patients with depression, and if these changes could reflect changes in the peripheral levels of members of the IGF family. To some extent, peripheral IGF proteins may be an interesting area to explore in large studies, evaluating not only the severity or symptomatology of depression, but also possible cognitive anomalies.

### Limitations and Future Perspectives

This study encountered the following limitations. First, even though we detected some statistical differences in IGFs proteins between controls and depressed patients, the underlying mechanism through which the IGF signaling system may be affecting depression is still not known. Second, changes in the peripheral protein levels might not represent changes in the central nervous system, or at least, this consideration was not within the scope of this study. Third, despite looking for possible correlations between metabolic parameters such as glucose, albumin, cholesterol, and triglyceride levels and IGFs proteins in the depressed group of patients, we did not have access to the levels of these metabolic parameters in controls or in the group of depressed patients after treatment, precluding us from forming an illustrative contrast. This would have been helpful in revealing whether these parameters changed in response to AD treatment. In general terms, more studies with bigger sample sizes are needed to reveal the connections between the IGF signaling system, cognitive state, and depression, and to find out whether the IGF family could be of use as potential biomarkers.

## 4. Materials and Methods

### 4.1. Experimental Design

We present a cross-sectional and partially longitudinal observational study that begun in October 2017 and finished in January 2022. We recruited 51 patients who met the DSM-V diagnostic criteria for depression (D group, n = 51) at the Álvaro Cunqueiro Hospital (Vigo, Spain). We also recruited 48 individuals without any previous psychiatric conditions that were included as controls (C group, n = 48). Among the 51 depressed patients, we recruited 15 (D0 and D1 groups, n = 15) patients after a period of 19 ± 6 days of treatment.

The inclusion criteria included meeting the DSM-V depression diagnostic criteria, age equal to or above eighteen years old (≥18 years), and the proper delivery of signed written consent. The exclusion criteria included additional neurological pathologies or other diseases that could interfere with our study, such as viral infections (COVID-19), cancer or cardiovascular disease. We have also excluded women with conditions such as pregnancy or lactation.

All patients and controls that participated in this study had Spanish nationality. We carried out this research according to the requirements of the Declaration of Helsinki and, so we were given the corresponding ethical approval (Code 2018/598). We have also obtained written consent from all patients and controls, or their corresponding legal guardians if needed.

### 4.2. Blood Collection and Plasma Obtention

Blood collection in EDTA tubes took place in the morning after a fasting period. Plasma was immediately separated in a Ficoll–Paque (3 mL 1:1 blood) gradient by centrifugation (2000 rpm, 35 min) and then aliquoted and stored in the freezer (−80 °C) until protein measurement.

### 4.3. Subjective Scales

We measured the severity of depression using the Hamilton Depression Rating Scale (HDRS) (24-item version) [69]. Additionally, we used the Self-Assessment Anhedonia Scale (SAAS) [96] in order to measure anshedonia among depressed patients and controls. Briefly, the scale measures 3 different domains (physical, social and intellectual) and it is composed of 27 items, each with 3 possible answers (intensity, frequency and change) rated on a Likert-like scale from 0 to 10. The possible score in each domain ranges from 0 to 270 points for a maximum of 810 points.

The general cognitive state was assessed using the Mini-Mental State Examination (MMSE) [97]. On the other hand, we used the Free and Cued Selective Reminding Test (FCSRT) [98] to assess episodic memory. Briefly, the FCSRT distinguishes between immediate and delayed free-recall and cued-facilitated immediate and delayed recall. Patients were shown a card with four words, and were then asked to identify which one of the four corresponds to a specific category (e.g., cue/category; “bird”, and the word belonging to that category was “raven”, not “car”, “vest” or “lemon”). Patients had to learn the four words or items on the four cards for a total of 16 words. Then, three recall trials were conducted, each one preceded by 20 s of counting as interference. In each trial, patients were asked to freely recall (the original order of the words was not required) as many words as possible, and the category cues were given whenever the patient was not able to remember a word. The same procedure of recalling (freely and cued) was repeated after 30 min. The measures assessed were the following: total free recall (TFR; cumulative sum of free recall from the three trials; range 0–48), total recall (TR; cumulative sum of free recall plus cued recall from the three trials; range 0–48), delayed free recall (DFR; free delayed recall; range 0–16), and finally delayed total recall (DTR; free delayed recall plus cued delayed recall; range 0–16).

All patients (n = 51) underwent the same assessment, including after treatment (n = 15).

### 4.4. Metabolic Parameters

Regular blood tests used in clinics were performed on depressed patients to measure fasting glucose levels (mg/dL), albumin (g/dL), total cholesterol (mg/dL) and total triglycerides (mg/dL) at the point of hospitalization, and this parameter was used to evaluate its correlation with the circulating IGF proteins.

### 4.5. Plasma Protein Measurement

First, we slowly defrosted plasma samples. Then, we measured the levels of IGF-2 (Catalog N° EH0166), IGFBP-1 (EH0167), IGFBP-3 (EH0169), IGFBP-5 (EH0405) and IGFBP-7 (EH0171) plasma proteins using an ELISA available commercial kit (Wuhan, China. Fine Biotech Co., Ltd.).

The process is briefly described elsewhere [32], and follows the instructions given by the manufacturers. However, in this protocol, we subtracted the value yielded by the 570 nm wavelength spectrophotometry measurement from that yielded by the 450 nm one in order to avoid or reduce possible plastic or residue contamination caused by the plate.

The intra-assay CVs (%), analytical sensitivity (M ± 3 SD) and reference range (min–max) were as follow: IGF-2 (6.15%, 59.92 ± 12.44 pg/mL and 62.5–4000 pg/mL, respectively), IGFBP-1 (7.59%, 0.145 ± 0.06 ng/mL and 0.156–10 ng/mL), IGFBP-3 (5.18%, 0.23 ± 0.11 ng/mL and 0.391–25 ng/mL), IGFBP-5 (4.03%, 47.77 ± 6.93 pg/mL and 62.5–4000 pg/mL) and IGFBP-7 (6.64%, 35.75 ± 4.52 pg/mL and 31.25–2000 pg/mL). All CVs were below 8%, which is the value recommended by the manufacturer.

### 4.6. Statistical Analysis

The quantitative data are shown in the form of mean and standard deviation (M ± SD) for each parameter; we used either the Shapiro–Wilk test or the Smirnov–Kolmogorov test to check whether quantitative parameters (IGF-2, IGFBP-1, IGFBP-3, IGFBP-5, IGFBP-7, HDRS, MMSE, SAAS, FSCRT, glucose, albumin, cholesterol, triglycerides, and age) could be adjusted to a normal distribution (reported as: S-W or S-K (df) = F, *p* > 0.05) or not. If these parameters were normally distributed and positive according to the Levene’s test, we used a parametric Student’s *t* test (reported as: *t* (df) = F, *p*-value). If not, we used a non-parametric test, such as Mann–Whitney (reported as: U, *p*-value). When comparing longitudinal data such as IGF protein levels before and after treatment, we used a parametric paired *t* test or a non-parametric paired test such as Wilcoxon’s test (reported as: W (df) = W, *p*-value). On the other hand, we used either Pearson’s correlation coefficient (reported as: r_p_ (df) = r_p_, *p*-value) (if both variables followed a normal distribution) or Spearman’s correlation coefficient (reported as: r_s_ (df) = r_s_, *p*-value) (if not), and we performed a multiple linear regression analysis via the backpropagation method to predict the effects of different independent variables (group, age and gender) on the values of dependent variables (IGF-2, IGFBP-1, IGFBP-3, IGFBP-5, IGFBP-7). We used the software GraphPad Prism 7.05 version. 

## 5. Conclusions

In this study, we have found that the plasma levels of IGF-2 and IGFBP-7 are significantly increased in depressed patients. However, only the level of IGF-2 remained significant after correction by age and sex. Moreover, the levels of IGF-2, IGFBP-3 and IGFBP-5 were significantly decreased after a period of treatment with antidepressants. On the other hand, IGFBP-3 and IGFBP-5 were found to be positively correlated with cognition and memory in depressed patients. These peripheral changes may reflect changes at the brain level, and could thus not only help us to understand the inner mechanisms of depression, but also act as a way of finding objective biomarkers that could help clinicians in daily practice. From our perspective, the neurotropic hypothesis, which suggests the reduced expressions of these factors as the mechanism behind depression, might offer an interesting framework to approach future research. More studies evaluating peripheral members of the IGF signaling system might be helpful in elucidating the potential roles of these proteins as possible biomarkers in depression.

## Figures and Tables

**Figure 1 ijms-24-15254-f001:**
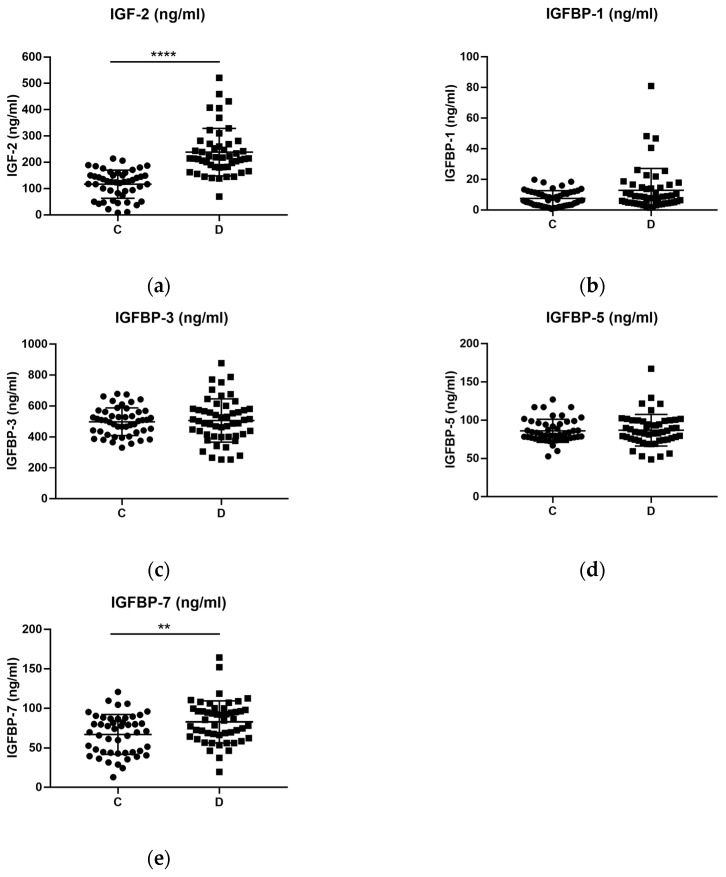
Plasma levels of IGF (ng/mL) proteins in healthy controls (C) and depressed patients (D). (**a**) Levels of IGF-2 were significantly increased in the depressed group when compared to the healthy control group with a Mann–Whitney unpaired test. In the cases of (**b**) IGFBP-1, (**c**) IGFBP-3 and (**d**) IGFBP-5, there were no significant differences in plasma levels between the controls and the depressed patients. (**e**) Levels of IGFBP-7 were also significantly increased in the depressed group after a Student’s *t* unpaired test. C: Control group. D: Depressed group of patients. ** *p*-value ≤ 0.01; **** *p*-value ≤ 0.0001.

**Figure 2 ijms-24-15254-f002:**
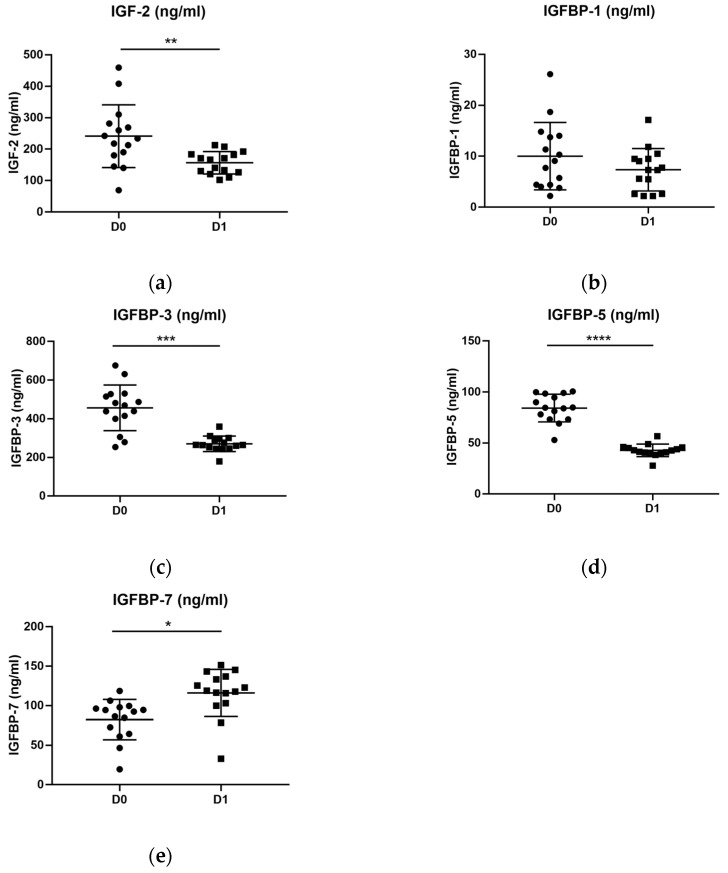
Plasma levels of IGFs (ng/mL) proteins in patients before (D0) and after (D1) a period of antidepressant treatment. We used a Wilcoxon paired test to compare both distributions. (**a**) Levels of IGF-2 were significantly decreased after treatment. In the case of (**b**) IGFBP-1, no statistical difference was detected. (**c**) IGFBP-3 and (**d**) IGFBP-5 also showed a significant reduction after treatment. (**e**) On the contrary, levels of IGFBP-7 were found to be significantly increased after treatment. D0: patients with depression before treatment. D1: patients with depression after treatment. * *p*-value ≤ 0.05; ** *p*-value ≤ 0.01; *** *p*-value ≤ 0.001; **** *p*-value ≤ 0.0001.

**Table 1 ijms-24-15254-t001:** Descriptive and general data from controls and depressed patients.

Variables	Control (n = 48)	Depression (n = 51)	*p*-Value
Gender (F/M)	22/26	30/21	0.230 ^1^
Age (years)	42.58 ± 11.54	52.71 ± 14.57	0.153 ^2^
SAAS (total)	108.17 ± 59.32	480.77 ± 160.62	<0.001 ^3^
IGF-2 (ng/mL)	116.69 ± 53.40	249.86 ± 119.57	<0.001 ^3^
IGFBP-1 (ng/mL)	7.57 ± 5.08	12.84 ± 14.31	0.061 ^3^
IGFBP-3 (ng/mL)	498.26 ± 89.57	506.12 ± 139.91	0.809 ^3^
IGFBP-5 (ng/mL)	86.14 ± 15.12	86.95 ± 20.69	0.904 ^3^
IGFBP-7 (ng/mL)	67.01 ± 25.14	82.90 ± 26.44	<0.01 ^4^

The mean values and the standard deviations of the different variables are shown. The *p*-value is shown in the fourth column, and is reported according to the APA style; ^1^ The Chi-square test was used to compare the distribution of sex between groups. ^2^ The median test for independent samples was used to compare the age distribution between both groups. ^3^ The Mann–Whitney test was employed to compare two continuous variables whenever at least one distribution did not adjust for a normal distribution. ^4^ Student’s *t* test was used. SAAS: Self-Assessment Anhedonia Scale. IGF: Insulin-like growth factor. IGFBP: Insulin-like growth factor-binding protein.

**Table 2 ijms-24-15254-t002:** Descriptive and general data from depressed patients before and after treatment.

D	D0 (n = 15)	D1 (n = 15)	*p*-Value
Gender (F/M)	7/8		-
Age (years)	61.60 ± 10.60		-
SAAS (total)	400.35 ± 160.41	370.40 ± 136.26	0.277 ^1^
HDRS	24.93 ± 6.18	15.00 ± 4.11	<0.001 ^2^
MMSE	23.53 ± 4.52	24.80 ± 5.45	<0.05 ^2^
FCSRT (TFR)	27.97 ± 10.81	31.49 ± 10.55	<0.01 ^1^
FCSRT (TR)	33.91 ± 9.12	35.71 ± 10.90	<0.05 ^1^
FCSRT (DFR)	10.43 ± 3.50	12.13 ± 3.55	<0.01 ^1^
FCSRT (DTR)	12.94 ± 3.46	14.20 ± 2.39	<0.01 ^1^
IGF-2 (ng/mL)	241.29 ± 99.86	156.71 ± 35.64	<0.01
IGFBP-1 (ng/mL)	10.02 ± 6.61	7.35 ± 4.16	0.258
IGFBP-3 (ng/mL)	456.49 ± 118.11	270.14 ± 39.86	<0.001
IGFBP-5 (ng/mL)	84.24 ± 13.56	42.78 ± 6.14	<0.001
IGFBP-7 (ng/mL)	82.36 ± 25.50	116.19 ± 29.85	<0.05

The mean values and the standard deviations of the different variables are shown. The *p*-value was calculated and is shown in the fourth column, and it is reported according to the APA style; ^1^ The Wilcoxon matched pairs signed rank test was used. ^2^ Student’s *t* test was used. D0: patients before treatment. D1: patients after treatment. SAAS: Self-Assessment Anhedonia Scale. HDRS: Hamilton Depression Rating Scale. MMSE: Mini-Mental State Examination. FCSRT: Facilitated and Cued Selective Reminding Test. TFR: Total Free Recall. TR: Total Recall. DFR: Delayed Free Recall. DTR: Delayed Total Recall. IGF: Insulin-like growth factor. IGFBP: Insulin-like growth factor-binding protein.

**Table 3 ijms-24-15254-t003:** Correlation between IGFs proteins and metabolic parameters in depressive patients (D).

Metabolic Parameters		IGF Plasma Proteins				
		IGF-2 (ng/mL)	IGFBP-1 (ng/mL)	IGFBP-3 (ng/mL)	IGFBP-5 (ng/mL)	IGFBP-7 (ng/mL)
Glucose (mg/dL)	r	0.105	0.231	0.035	−0.004	0.085
	*p*	0.469	0.103	0.805	0.977	0.522
Albumin (g/dL)	r	−0.121	0.228	−0.115 ^p^	−0.216	−0.066 ^p^
	*p*	0.407	0.111	0.425	0.131	0.648
Cholesterol (mg/dL)	r	0.155	0.190	0.158 ^p^	0.192	0.168 ^p^
	*p*	0.293	0.195	0.284	0.190	0.252
Triglycerides (mg/dL)	r	0.138	0.268	−0.108	−0.003	0.275 ^p^
	*p*	0.349	0.065	0.464	0.983	0.058

Spearman’s correlation coefficient was the main correlation coefficient employed. r: Coefficient values range from 1 to −1. ^p^ Pearson’s correlation coefficient was employed when both distributions followed a normal distribution (Shapiro–Wilk; *p* > 0.05). IGF: Insulin-like growth factor. IGFBP: Insulin-like growth factor-binding protein.

**Table 4 ijms-24-15254-t004:** Linear regression models for the effects of group, age and gender over IGF-2 levels.

Predictors	F (df1, df2)	R^2^	*p*	β	SD	*p^β^*
Model 1	17.35 (3, 94)	0.36	<0.001	71.96	36.28	0.05
Group				121.74	20.62	<0.001
Age				0.83	0.73	0.258
Gender				20.45	19.33	0.293
Model 2	25.44 (2, 95)	0.35	<0.001	86.86	33.45	0.011
Group				125.97	20.24	<0.001
Age				0.83	0.73	0.332
Model 3	49.96 (1, 96)	0.34	<0.001	116.69	13.45	<0.001
Group				133.17	18.84	<0.001

Linear regression models were performed using the backpropagation method, excluding variables from the model that do not have a significant impact on the variance explained. Df1: degrees of freedom of the independent variables. df2: degrees of freedom of the dependent variable. R^2^: R-squared. *p*: *p*-value of the regression equation of each model. β: beta coefficients. SD: Standard deviation of beta coefficients. *p^β^* = *p*-value of the beta coefficients. Group is a dichotomic variable formed from mental condition (healthy, depression). Age is a continuous variable. Gender is a dichotomic variable formed from sex (male, female). We show exact *p*-values for the better understanding of the models.

**Table 5 ijms-24-15254-t005:** Linear regression models for the effects of group, age and gender over IGFBP-7 levels.

Predictors	F (df1, df2)	R^2^	*p*	β	SD	*p^β^*
Model 1	9.37 (3, 95)	0.23	<0.001	31.18	9.29	0.001
Group				7.21	5.26	0.174
Age				0.76	0.19	<0.001
Gender				7.92	4.93	0.112
Model 2	13.00 (2, 96)	0.21	<0.001	29.61	9.26	0.002
Age				0.85	0.17	<0.001
Gender				9.13	4.88	0.064

Linear regression models were performed using the backpropagation method, excluding variables from the model that do not have a significant impact on the variance explained. Df1: degrees of freedom of the independent variables. df2: degrees of freedom of the dependent variable. R^2^: R-squared. *p*: *p*-value of the regression equation of each model. β: beta coefficients. SD: Standard deviation of beta coefficients; *p^β^* = *p*-value of the beta coefficients. Group is a dichotomic variable formed from mental condition (healthy, depression). Age is a continuous variable. Gender is a dichotomic variable formed from sex (male, female). We show exact *p*-values for the better understanding of the models.

**Table 6 ijms-24-15254-t006:** Correlation between IGFs proteins and subjective scales in depressed patients before treatment.

Subjective Scales		IGF Plasma Proteins				
		IGF-2 (ng/mL)	IGFBP-1 (ng/mL)	IGFBP-3 (ng/mL)	IGFBP-5 (ng/mL)	IGFBP-7 (ng/mL)
HDRS	r	−0.437	−0.077	−0.002	0.007	−0.447
	*p*	0.104	0.784	0.995	0.980	0.104
MMSE	r	−0.229	<0.000	0.714	0.670	0.155
	*p*	0.104	0.999	0.003	0.006	0.581
SAAS	r	−0.289	0.250	−0.064	−0.068	−0.089
	*p*	0.296	0.369	0.820	0.810	0.752
FCSRT_TFR	r	−0.256	−0.066	0.556	0.456	−0.077
	*p*	0.358	0.815	0.031	0.088	0.785
FCSRT_TR	r	−0.298	−0.034	0.754	0.672	−0.005
	*p*	0.28	0.904	0.001	0.006	0.985
FCSRT_DFR	r	−0.125	−0.05	0.409	0.294	0.183
	*p*	0.656	0.859	0.130	0.288	0.514
FCSRT_DTR	r	−0.394	−0.027	0.486	0.452	−0.013
	*p*	0.146	0.924	0.066	0.091	0.964

Spearman’s correlation coefficient, r: Coefficient values range from 1 to −1. Pearson’s correlation coefficient was employed when data followed a normal distribution (Shapiro–Wilk; *p* > 0.05). SAAS: Self-Assessment Anhedonia Scale. HDRS: Hamilton Depression Rating Scale. MMSE: Mini-Mental State Examination. FCSRT: Facilitated and Cued Selective Reminding Test. TFR: Total Free Recall. TR: Total Recall. DFR: Delayed Free Recall. DTR: Delayed Total Recall. IGF: Insulin-like growth factor. IGFBP: Insulin-like growth factor-binding protein.

**Table 7 ijms-24-15254-t007:** Correlation between IGFs proteins and subjective scales in depressed patients after treatment.

Subjective Scales		IGF Plasma Proteins				
		IGF-2 (ng/mL)	IGFBP-1 (ng/mL)	IGFBP-3 (ng/mL)	IGFBP-5 (ng/mL)	IGFBP-7 (ng/mL)
HDRS	r	−0.183	0.457	−0.120	0.045	0.219
	*p*	0.514	0.087	0.669	0.874	0.433
MMSE	r	−0.287	0.028	0.285	0.66	−0.287
	*p*	0.3	0.921	0.303	0.007	0.015
SAAS	r	−0.372	0.309	−0.252	−0.239	−0.084
	*p*	0.172	0.263	0.365	0.390	0.766
FCSRT_TFR	r	−0.206	0.068	0.493	0.608	−0.422
	*p*	0.461	0.809	0.062	0.016	0.188
FCSRT_TR	r	−0.251	0.295	0.410	0.566	−0.380
	*p*	0.368	0.286	0.129	0.028	0.163
FCSRT_DFR	r	−0.203	−0.129	0.161	0.483	−0.322
	*p*	0.469	0.647	0.566	0.068	0.242
FCSRT_DTR	r	−0.286	−0.162	0.099	0.429	−0.3
	*p*	0.302	0.565	0.726	0.111	0.277

Spearman’s correlation coefficient, r: Coefficient values range from 1 to −1. Pearson’s correlation coefficient was employed when data followed a normal distribution (Shapiro–Wilk; *p* > 0.05). SAAS: Self-Assessment Anhedonia Scale. HDRS: Hamilton Depression Rating Scale. MMSE: Mini-Mental State Examination. FCSRT: Facilitated and Cued Selective Reminding Test. TFR: Total Free Recall. TR: Total Recall. DFR: Delayed Free Recall. DTR: Delayed Total Recall. IGF: Insulin-like growth factor. IGFBP: Insulin-like growth factor binding protein.

## Data Availability

The data that support the findings of this study are available from the corresponding author upon reasonable request.

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
