# Peer review of "Protein Plasma Levels of the IGF Signalling System Are Altered in Major Depressive Disorder"

_ijms, 2023, doi:10.3390/ijms242015254_

Round 1

Reviewer 1 Report

MDPI

Manuscript: IJMS-2614466

In this manuscript, the authors studied protein plasma levels of the IGF signaling system are altered in major depressive disorder. The authors presented descriptive and general data from control and depressed patient (Table 1), correlation between IGFs proteins and metabolic parameters in depressed patients (Table 2), descriptive and general data from depressed patients before and after treatment (Table 3), linear regression models for the effect of group, age and gender over IGF-2 levels (Table 4), linear regression models for the effect of group, age and gender over IGFBP-7 level (Table 5), plasma levels of IGF in healthy controls and depressed patients (Fig. 1), plasma levels of IGFs in patients before and after antidepressant treatment (Fig. 2),  correlation between IGFs proteins and subjective scale in depressed patients before treatments (Table 6), and after treatment (Table 7). The authors concluded and reported Both IGF-2 and IGFBP-7 levels were found significantly increased in the depressed groups, and the IGF-2, IGFBP-3 and IGFBP-5 levels were significantly depressed after treatment, whereas IGFBP-7 was significantly increased. The authors have provided novel information in the protein levels of IGF signaling system of major depressive disorder.

Author Response

Rebuttal- Round 1

Reviewer 1 

In this manuscript, the authors studied protein plasma levels of the IGF signaling system are altered in major depressive disorder. The authors presented descriptive and general data from control and depressed patient (Table 1), correlation between IGFs proteins and metabolic parameters in depressed patients (Table 2), descriptive and general data from depressed patients before and after treatment (Table 3), linear regression models for the effect of group, age and gender over IGF-2 levels (Table 4), linear regression models for the effect of group, age and gender over IGFBP-7 level (Table 5), plasma levels of IGF in healthy controls and depressed patients (Fig. 1), plasma levels of IGFs in patients before and after antidepressant treatment (Fig. 2),  correlation between IGFs proteins and subjective scale in depressed patients before treatments (Table 6), and after treatment (Table 7). The authors concluded and reported Both IGF-2 and IGFBP-7 levels were found significantly increased in the depressed groups, and the IGF-2, IGFBP-3 and IGFBP-5 levels were significantly depressed after treatment, whereas IGFBP-7 was significantly increased. The authors have provided novel information in the protein levels of IGF signaling system of major depressive disorder.

Dear Reviewer 1, 

We would like to thank you for your comments and report, but also for the recognition of our work. 

Kind regards,

Roberto C. Agís-Balboa & Carlos Fernandez-Pereira 

Reviewer 2 Report

Above all, the paper is prepared extremely carefully. The study provides new information on the biological basis of depression, including the IGF signaling system.

Nevertheless, I have the following comments:

Study period 1 covers the COVID-19 pandemic - were such patients included in the study?

2. ELISA kits must be thoroughly characterized - CVs, detection limits, and reference ranges.

Author Response

Reviewer 2

Comments and Suggestions for Authors

Above all, the paper is prepared extremely carefully. The study provides new information on the biological basis of depression, including the IGF signaling system.

Nevertheless, I have the following comments:

Study period 1 covers the COVID-19 pandemic - were such patients included in the study?

  1. ELISA kits must be thoroughly characterized - CVs, detection limits, and reference ranges.

Dear Reviewer 2,

Thank you for your comments and suggestions that will definitively help to improve the exposition of our work.

As we state in point “4. Materials and Methods; 4.1 Experimental Design” (lines 426-428) we included in our procedure several exclusion criteria. We did not directly mention that COVID-19 was part of a bigger exclusion criteria but we supposed it under the statement “or other diseases that could interfere with our study” (lines 426-427). However, we will now add to that sentence: “such as viral infections (COVID-19)”.

Indeed, we have added a paragraph in point “4. Materials and Methods; 4.5 Protein Measurement” in which we characterized the main parameters in the ELISA kits (lines 485-489) “The intra-assay CVs (%), analytical sensitivity (M ± 3 SD) and reference range (min-max) were as follow: IGF-2 (6.15%, 59.92 ± 12.44 pg/ml and 62.5-4000 pg/ml), IGFBP-1 (7.59%, 0.145 ± 0.06 ng/ml and 0.156-10 ng/ml), IGFBP-3 (5.18%, 0.23 ± 0.11 ng/ml and 0.391-25 ng/ml), IGFBP-5 (4.03%, 47.77 ± 6.93 ng/ml and 62.5-4000 pg/ml) and IGFBP-7 (6.64%, 35.75 ± 4.52 ng/ml and 31.25-2000 pg/ml). All CVs were below 8% which is recommended by the manufacturer.”

We have calculated the CV for each sample extracted from each control or patient as the standard deviation  divided by the mean value  of both replicates (or more).                         

Then,

We calculated the average CV for each ELISA KIT (IGF-2, IGFBP-1, IGFBP-3, IGFBP-5 and IGFBP-7). We obtained the following CVs: IGF-2 (6.15%). IGFBP-1 (7.59%), IGFBP-3 (5.18%), IGFBP-5 (4.03%) and IGFBP-7 (6.64%).

Functional sensitivity or the limit of lowest quantification (LLOQ) of the Kits were given by the manufacturers: IGF-2 (37.5 pg/ml), IGFBP-1 (0.094 ng/ml), IGFBP-3 (0.234 ng/ml), IGFBP-5 (37.5 pg/ml), IGFBP-7 (18.75 pg/ml) and range detections: IGF-2 (62.5-4000 pg/ml), IGFBP-1 (0.156-10 ng/ml), IGFBP-3 (0.391-25 ng/ml), IGFBP-5 (62.5-4000 pg/ml), IGFBP-7 (31.25-2000 pg/ml). In terms of analytical sensitivity or limit of detection (LOD) that is the lowest analyte concentration that can be detected from the background was calculated as the background ± 3 SD. According to Shinkai (1996) this is a valid, among other three methods, to calculate analytical sensitivity. Analytical sensitivity was as follow: IGF-2 (59.92 ± 12.44 pg/ml), IGFBP-1 (0.145 ± 0.06 ng/ml), IGFBP-3 (0.23 ± 0.11 ng/ml), IGFBP-5 (47.77 ± 6.93 pg/ml) and IGFBP-7 (35.75 ± 4.52 pg/ml).

Thank you very much again for your comments and suggestions and we hope the quality of our work had improved.

Kind regards,

The scientific team of Neuro Epigenetics Lab.

References

Shinkai E. [Comparison of methods to determine analytical sensitivity]. Rinsho Byori. 1996 Nov;44(11):1067-71. Japanese. PMID: 8953937.

Reviewer 3 Report

Although it may have good contributions to some medical fields, contributions to molecular sciences are limited. The aim of this research is to assess alterations in peripheral proteins belonging to the Insulin-like growth factor family in depression. For this purpose, the data were collected from patients (a depressed group of patients (N = 51) and a healthy control group (N = 48)). The most of data were provided only as values, but any evidences on molecular analyses were not provided. This way of research would not be effective in investigation for molecular sciences. I do not think that IJMS is appropriate journal for this work. It is better to change publication journal.

Author Response

Reviewer 3

Comments and Suggestions for Authors

Although it may have good contributions to some medical fields, contributions to molecular sciences are limited. The aim of this research is to assess alterations in peripheral proteins belonging to the Insulin-like growth factor family in depression. For this purpose, the data were collected from patients (a depressed group of patients (N = 51) and a healthy control group (N = 48)). The most of data were provided only as values, but any evidences on molecular analyses were not provided. This way of research would not be effective in investigation for molecular sciences. I do not think that IJMS is appropriate journal for this work. It is better to change publication journal.

Dear Reviewer 3,

We would like to thank you for your comments and suggestions.

To begin with, we partially agree on that the research design could be improved as we state in the Limitations and Future Perspectives topic (lines 410-415). “Third, despite of looking for possible correlations between metabolic parameters such as glucose, albumin, cholesterol, and triglycerides levels and IGFs proteins in the depressed group of patients, we did not have access to the levels of these metabolic parameters in controls in order to make a contrast, neither in the group of depressed patients after treatment. The last could have been interesting to check whether these parameters might have changed in response to antidepressant treatment”.

We thought that maybe adding the exact p-values could be interesting to improve the exposition of our work. Initially, we did not make it because we were following the APA style notation that recommends to give non-significant p-values as “ns” (non-significant) in charts. We have also followed your recommendations and we have modified the conclusion in order to make it more suitable.

We decided to send our work to this journal and more in specific, to this special issue entitled “The Role of the IGF Axis in Disease 3.0” because we are indeed studying the role of some members of the IGF signalling system in the context of a disease that is depression, a psychiatric disorder.

Therefore, the main problematic with psychiatric disorders and several other neurodegenerative or brain-related diseases is that we cannot have access to the brain itself. We cannot measure or study molecular anomalies in that sense, neither gene expression or protein levels, post-translational alterations, etc. For example, if we wanted to study anomalies in the expression of members of the IGF signalling members in the brain we can only do it by using post-mortem samples. So, as it is stated in an article published in IJMS (special issue: Schizophrenia: Pathophysiology, Diagnostics, Therapies, and Prevention) “Biomarkers of schizophrenia may be divided into peripheral and central biomarkers. However, since schizophrenia is a disease of systemic nature, some biomarkers (“i.e., analytes from the post mortem brains of patients with schizophrenia) are related to changes found in the blood, suggesting that brain/CNS and periphery are interconnected and therefore blood-based biomarkers are useful tools to reveal some processes in the brain”)” (Perkovic et al 2017). The same could be applied to depression as a psychiatry disorder. We give this example to show how variable this could be and that is not just closed to depression but to other mental conditions.

The point I wanted to prove is that, although we are not studying direct molecular changes in the proteins themselves, we are measuring the way the expression of these proteins could be altered in the brain, and the way that can be measured and reported as a peripheral biomarker in plasma. This is why we deeply consider that our work is suitable for this journal.

On the other hand, there are several articles that are addressing similar issues in IJMS like a recent systematic review titled “Brain-Derived Neurotrophic Factor (BDNF) as a Predictor of Treatment Response in Major Depressive Disorder (MDD): A Systematic Review” (Zelada et al 2023) or an original article like “Antidepressant Medication Does Not Contribute to the Elevated Circulating Concentrations of Acylethanolamides Found in Substance Use Disorder Patients” (Herrera-Imbrued et al 2023).

Moreover, our research team have already published a paper in this journal with a similar approach back on 2022 (Fernández-Pereira et al 2022).

Kind regards,

Roberto C. Agís-Balboa & Carlos Fernández-Pereira

References.

Perkovic, M.N.; Erjavec, G.N.; Strac, D.S.; Uzun, S.; Kozumplik, O.; Pivac, N. Theranostic Biomarkers for Schizophrenia. Int. J. Mol. Sci. 2017, 18, 733. https://doi.org/10.3390/ijms18040733

Zelada, M.I.; Garrido, V.; Liberona, A.; Jones, N.; Zúñiga, K.; Silva, H.; Nieto, R.R. Brain-Derived Neurotrophic Factor (BDNF) as a Predictor of Treatment Response in Major Depressive Disorder (MDD): A Systematic Review. Int. J. Mol. Sci. 2023, 24, 14810. https://doi.org/10.3390/ijms241914810

Herrera-Imbroda, J.; Flores-López, M.; Requena-Ocaña, N.; Araos, P.; García-Marchena, N.; Ropero, J.; Bordallo, A.; Suarez, J.; Pavón-Morón, F.J.; Serrano, A.; et al. Antidepressant Medication Does Not Contribute to the Elevated Circulating Concentrations of Acylethanolamides Found in Substance Use Disorder Patients. Int. J. Mol. Sci. 2023, 24, 14788. https://doi.org/10.3390/ijms241914788

Fernández-Pereira C, Penedo MA, Rivera-Baltanas T, Fernández-Martínez R, Ortolano S, Olivares JM, Agís-Balboa RC. Insulin-like Growth Factor 2 (IGF-2) and Insulin-like Growth Factor Binding Protein 7 (IGFBP-7) Are Upregulated after Atypical Antipsychotics in Spanish Schizophrenia Patients. Int J Mol Sci. 2022 Aug 24;23(17):9591. doi: 10.3390/ijms23179591. PMID: 36076984; PMCID: PMC9455262.

Reviewer 4 Report

The authors measured plasma IGF-2, IGFBP-1, IGFBP-3, IGFBP-5, IGFBP-7 protein levels in a depressed group of patients (N = 51) and in a healthy control group (N = 48).

Minor:

-abstract: conclusion is missing.

-non significant values should be also included in tables.

-first sentence in conclusion part in unnecessary (lines 502-504).

Author Response

Reviewer 4

Comments and Suggestions for Authors

The authors measured plasma IGF-2, IGFBP-1, IGFBP-3, IGFBP-5, IGFBP-7 protein levels in a depressed group of patients (N = 51) and in a healthy control group (N = 48).

Dear Reviewer 3,

First, thank you for reviewing our work and giving us advices so we can improve it.

We have added the exact values of non-significant p-values in all tables. We did not do it initially because we were following the APA style notation that is normally used in figures (graphs and charts) in order to facilitate the identification of significant p-values. However, we did not apply that rule for Tables 4 and 5 in which we did several correlation analyses to avoid redundance of “ns”.

In brief, thank you for noticing it because we agree on that it will help to improve the exposition of our work.

On the other hand, we have also added a super-index indicating the type of statistical test that was used in each row of table 3.

We have also changed the conclusion in order to give a more explained panorama of the situation of the IGF signalling system in our study and in general terms linked to depression. We have changed the conclusion to (lines 509-520) “In this study, we have found that plasma levels of IGF-2 and IGFBP-7 are significantly increased in depressed patients. However, only IGF-2 remained significant after correcting by age and sex. Moreover, IGF-2, IGFBP-3 and IGFBP-5 significantly de-creased after a period of treatment with antidepressants. On the other hand, IGFBP-3 and IGFBP-5 have been found positively corelated with cognition and memory in de-pressed patients. These peripheral changes may be recreating changes at brain level that could help to understand not only inner mechanisms of depression but also is a way of finding objective biomarkers that could help clinicians in daily practice. From our perspective, the neurotropic hypothesis that suggests that reduced expression of these factors could be a mechanism behind depression might be an interesting approach in future research. More studies evaluating peripheral members of the IGF signaling system might be interesting discuss the potential role of these proteins as possible biomarkers in depression.”

Minor:

-abstract: conclusion is missing

We have changed the abstract to

The Insulin-like growth factor 2 (IGF-2) has been recently proved to alleviate depressive-like behaviors in both rats and mice models. However, its potential role as a peripheral biomarker has not been evaluated in depression. In order to do this, we measured plasma IGF-2 and other members of the IGF family such as Binding Proteins: IGFBP-1, IGFBP-3, IGFBP-5 and IGFBP-7 in a depressed group of patients (N = 51) and in a healthy control group (N = 48). In some of these patients (N = 15) we measured these proteins after a period (19 ± 6 days) of treatment with anti-depressants. The Hamilton Depressive Rating Scale (HDRS) and the Self-Assessment Anhedonia Scale (SAAS) were used to measure depression severity and anhedonia, respectively. The general cognition state was assessed by the Mini-Mental State Examination (MMSE) test and memory with the Free and Cued Selective Reminding Test (FCSRT). The levels of both IGF-2 and IGFBP-7 were found significantly increased in the depressed group, however only IGF-2 remained significant after correction by age and sex. On the other hand, the levels of IGF-2, IGFBP-3 and IGFBP-5 were significantly decreased after treatment, whereas only IGFBP-7 was significantly increased. Therefore, peripheral changes in the IGF family and their response to antidepressants might be representing alterations at brain level in depression.”.

-non significant values should be also included in tables.

We have now included the exact p-values of all non-significant values in all tables.  

-first sentence in conclusion part in unnecessary (lines 502-504).

We included those lines (“The Insulin-like Growth Factor family has been related with psychiatric disorders such as depression, schizophrenia, or bipolar disorder and with neurodegenerative disorders that alters the cognitive state like Alzheimer’s disease.”) to give a quick glimpse of the implication of the IGF family not only in psychiatry disorders but also in some neurodegenerative diseases such as Alzheimer’s disease. However, we agree on that is not a direct part (or a conclusion itself) of our study so we have already removed it.

Finally, we want to thank you for your comments and observations.

Kind regards,

Roberto C. Agís-Balboa & Carlos Fernández-Pereira

Round 2

Reviewer 2 Report

The authors have satisfactorily responded to all my questions and made the necessary changes to the manuscript.

Reviewer 3 Report

OK